# Towards Emotionally Expressive Virtual Human Agents to Foster L2 Production: Insights from a Preliminary Woz Experiment

**Emmanuel Ayedoun *** and **Masataka Tokumaru**

Faculty of Engineering Science, Kansai University, 3-3-35 Yamatecho, Suita, Osaka 564-8680, Japan
* Correspondence: emay@kansai-u.ac.jp

**Abstract:** In second-language communication, emotional feedbacks play a preponderant role in instilling positive emotions and thereby facilitating the production of the target language by second-language learners. In contrast, facial expressions help convey emotion, intent, and sometimes even desired actions more effectively. Additionally, according to the facial feedback hypothesis, a major component of several contemporary theories of emotion, facial expressions can regulate emotional behavior and experience. The aim of this study was to determine whether and to what extent emotional expressions reproduced by virtual agents could provide empathetic support to second-language learners during communication tasks. To do so, using the Facial Coding Action System, we implemented a prototype virtual agent that can display a collection of nonverbal feedbacks, including Ekman' six basic universal emotions and gazing and nodding behaviors. Then, we designed a Wizard of Oz experiment in which second-language learners were assigned independent speaking tasks with a virtual agent. In this paper, we outline our proposed method and report on an initial experimental evaluation which validated the meaningfulness of our approach. Moreover, we present our next steps for improving the system and validating its usefulness through large-scale experiments.

**Keywords:** virtual human agents; facial feedbacks; second-language communication



## 1. Introduction

Emotion or affect has long been unfairly deemed to be nothing than an insignificant byproduct of biological reasoning process, or even a destructive obstacle to controlled, logical reasoning and intelligent behavior [1]. Interestingly, views on the role and utility of emotion have changed remarkably over time, and its pervasiveness in human life in general and human communication in particular is now widely acknowledged. The human face is a prolific source of information about human emotional and behavioral states. Face expressions, nodding, and eye gazing are example of nonverbal behaviors that often accompany and regulate interactions in communication settings [2]. For example, the facial feedback hypothesis [3], an important component of many modern theories of emotion, suggests that facial expressions have a causal relationship between emotional experience and behavioral regulation. Conversely, empathic interlocutors who provide the appropriate amount of emotional responses can facilitate the arousal of positive emotions in second-language learners and facilitate their production of the target language [4].

In addition, previous work suggests that because emotions are at the heart of our humanity, conversational agents endowed with the ability to effectively convey appropriate emotional responses greatly enhance the illusion of life [5].

In this study, we aim to investigate whether and through which mechanisms emotionally expressive conversational agents that mimic non-verbal attentive behavior may display empathetic support to second-language learners during independent speaking tasks. To this end, we propose a method for generating facial expressions that utilizes a facial coding action system [6]. Then, we implement a prototype virtual agent that can display a set of non-verbal feedbacks or backchannels including Ekman' six basic universal emotions [7] in

addition to gazing and nodding behaviors. Here, we present a preliminary assessment of the significance of our approach towards eliciting second language communication.

The following is an overview of the remainder of this paper: we begin by situating the context of this work by referring to past and present works that are related to the key concepts of the present contribution. Then, we present an outline of the novelty and main objectives of our work. Next, we provide an overview of the proposed methods for enabling dynamic generation of emotional facial expressions by the virtual agent. In the following section, we describe the experimental study and its preliminary results. The final section of the paper presents some conclusions and discusses directions for future work.

## 2. Literature Review

### 2.1. Emotion and Second Language Production

According to Piaget [8], and as elaborated by Langer [9], cognition provides the organizational focus, while emotion is the energetic force of human activity. Piaget conceives of emotion as the energy source upon which the good functioning of cognition depends, as in a car where gas is necessary to make the motor run but does not affect the design of the motor. Thus, emotion and cognition are both complementary aspects of human activity. Learning is especially a human activity where emotional factors play a central role.

For instance, second-language (L2) learning has the fundamental goal of enabling learners to communicate effectively using that language when given the chance to do so. Swain [10] stipulated that the ability to produce written or oral output is also necessary for learners in addition to comprehensible input. The reason is that, as learners produce output, they have the opportunity to test their language hypotheses, receive corrective feedback on their incorrect productions, improve their metalinguistic awareness, and notice gaps between their interlanguage and the target language. However, it has been observed that many L2 learners will not spontaneously engage in a conversation in their second language despite studying it for several years. Prior research has suggested that the key factor ensuring frequency of L2 use in communicative situations is the willingness to communicate, defined as the "readiness to enter into discourse at a particular time with a specific person or persons, using an L2" [11]. Having found that learners with higher WTC perform better in terms of target language production than others, MacIntyre and his colleagues suggested that increasing learners' WTC should be the goal of L2 learning. Moreover, they proposed a pyramidal heuristic model of variables influencing WTC, in which it appears that situated emotional antecedents such as anxiety and self-confidence felt by second-language learners have a direct and substantial impact on their decision to participate (or not) in a conversation using the target language. Hence, the dialogue partner may play an important role in motivating the learner to communicate in L2, assuming that situated emotional antecedents are contingent on specific contexts in which people function at any given time. For instance, many learners feel a genuine anxiety about performing in front of others, and many classrooms do not, thus, offer learners much in the way of communicative practice, as would be desirable [12]. Isoda [13] mentioned that many Japanese university students hesitate to talk because they have little experience speaking English in Japan. It follows that increasing learners' ability to communicate in their second language is a problem that is difficult to overcome due to the lack of suitable conversation environments and partners.

Interestingly, there is some evidence that listener's behavior towards their interlocutors could have an impact on L2 speakers' fluency during oral tasks. Thus, those behaviors have received much deliberation [14]. When L2 speakers perform oral tasks, teachers or testers are often present and respond to the production with a variety of verbal and nonverbal messages. Among other descriptions, these messages have been described as "signals of attention" [15], "accompaniment signals" [16], "listener responses" [17], and "backchannels" [18] in the literature. It is generally admitted that, during conversation, people often convey information through two channels: the predominant or main channel, which is the channel through which speech flows, and the secondary, or backchannel, dedicated to the

transmission of meta-conversational signals [19]. In other terms, backchannels should be understood as a category of verbal or non-verbal expressions occurring on a conversation's secondary channel; they serve a meta-conversational purpose in the sense that they do not bring any new content-based information to the communication. Rather, they are used to support the primary speaker's turn by conveying the listener's comprehension and/or interest. The nonverbal emotional expressivity that we aim to achieve within virtual agents in the context of this study is intended to fulfill such meta-conversational purpose.

More recently, as suggested by Vo [20], interaction research has shifted its focus to examining the multidimensional construct of interaction and how the manipulation of these constructs affects L2 learning and acquisition. Further, a large body of research [21–25]) has recently explored how interactional components such as input, feedback, and output opportunities impact L2 development.

### 2.2. Animated Pedagogical Agents and Learning Support

Animated pedagogical agents are a particular type of embodied conversational agents that combine the pedagogical functions of intelligent tutoring systems with natural language dialogues. The social agency theory [26] outlines the effectiveness of such animated pedagogical agents in human–computer interaction. According to this theory, using verbal and visual cues in a computer-based environment encourages learners to interpret their interaction with the computer as a partnership. Learners consider their interaction with the computer a social one, because the social cues are similar to what they would expect from a human-to-human conversation.

Examples of successful animated pedagogical agents include AutoTutor [27], CIRCSIM-Tutor [28], Why2-Atlas [29], etc. Such systems foster deep learning as students are prompted to explain their reasoning and reflect on their problem-solving activities. Research in the field of computer-supported language learning has also been around for more than two decades, and several studies have actually yielded interesting results in terms of increasing learners' grammar, vocabulary, and reading learning skills, suggesting that computer-based learning environments could be an efficient alternative to real interactions. Unfortunately, such studies have essentially focused on cognitive aspects of learning, and affect or emotion-related phenomena have not been thoroughly addressed.

As far as second-language learning support is especially concerned, although animated pedagogical agents have been shown to offer L2 learners the opportunity to interact with "native speakers" and to provide a social context [30], computer-based learning support systems that target learners' engagement towards communication in particular remain a conspicuous rarity in the literature. Among the few attempts to propose a computer-based approach to increase levels of L2 learners' motivation towards the production of the target language [12,31,32], less effort has been expended on investigating the usage of realistic virtual interfaces, such as embodied conversational environments, which seem to have the potential to be a suitable alternative to face-to-face authentic interactions. There are, however, two rare exceptions, which have recently contributed to raising awareness of the benefits of computer-based dialogue agents towards supporting second language production. The first one is the Tactical Iraqi Language and Culture Training System [33]. Originally developed to provide US Marines with practical training on Iraqi culture, gestures, and situational language skills before being deployed on real-world missions, it is an advanced computer-agent-based software program. In the system, Marines must communicate face-to-face with animated characters representing local people through a virtual-world computer game that uses advanced artificial intelligence technologies. The second one is a more recent system: CEWill [34]. It is an embodied conversational agent that provides L2 learners with opportunities to naturally simulate daily conversations in various social contexts. The system was equipped with DiMaCA, a dialogue management model based on two verbal conversational strategies (i.e., communication strategies (CS) and affective backchannels (AB)) designed to foster L2 learners' willingness to communicate in an English-as-a-foreign-language context. It was found that the system could

yield interesting outcomes in terms of its ability to raise learners' engagement towards communication in their second language.

However, although these studies introduce unique new potentials and challenges from a pedagogical perspective, they seem to not focus enough on how to effectively manage the non-verbal, extra-linguistic dimension of the interaction between conversational agents and second language learners. As mentioned in the previous section, these "extra-linguistic" signals play a powerful role in defining the nature of social exchange. When these signals are positive, they can lead to feelings of rapport and promote beneficial outcomes in such diverse areas as negotiations and conflict resolution [35,36]. The need for such feedbacks in conversational agents is undeniable for reasons of naturalism or believability, as suggested by previous works which demonstrated the importance of backchannels during human–agent conversations, considering them an important milestone for building engaging and natural virtual humans ([37,38]). These works provide valuable insights on the idea that verbal but more importantly non-verbal backchannels may support emotional variables influencing L2 learners' motivation in a conversational agent-mediated interaction. In the same vein, Heckmann and Wobbroack suggest that conversational agents that can effectively convey appropriate emotional responses greatly augment the illusion of life because emotions are something that we find at the heart of what it means to be human [5].

### 2.3. Facial Actions Coding System

According to Ekman et al. [39], faces provide signals about how a person feels, which may influence other people's emotional experiences. A new method for measuring facial behavior, the Facial Action Coding System (FACS), was developed by Ekman and Friesen [40] in an effort to provide a sounder understanding of what different facial actions could reflect, and following on from the work of Hjortsjön [41]. FACS was primarily developed as a comprehensive system to distinguish all possible visible anatomically based facial movements. The system divides facial expressions into units of movement called Action Units. FACS provides a common nomenclature for facial movement research, which allows for diverse application in a variety of fields. Over the years, FACS has become the most widely exploited descriptive coding scheme for facial behaviors and is used in diverse fields such as neuroscience, computer vision, computer graphics and animation, facial encoding for digital signal processing, etc. [42]. Ekman et al. [43] later published a significant update to FACS. The FACS technique allows human observers to uniquely break down facial expressions into AUs, including nine action units in the upper face and 18 in the lower face. In addition, there are 14 head positions and movements, nine eye positions and movements, five miscellaneous action units, nine action descriptors (i.e., movements for which the anatomical basis is unspecified), nine gross behaviors, and five visibility codes. As it is believed that facial expressions and emotions are closely linked, FACS is commonly used for interpreting non-verbal communicating signals, such as facial expressions related to emotion or other human states [44]. For instance, as early as in 1987, testing Darwin's "universality hypothesis" from 1872, researchers Ekman & Friesen conducted a cross-cultural study in which respondents viewed pictures of human faces and were asked to identify which emotions were present on the faces, as well as rate how intense the emotions were [45]. The researchers reported that, even though minor cultural variations were present in the ratings of emotional intensity, all respondents were able to recognize which emotion was predominant and categorize it accordingly.

The facial feedback hypothesis also asserts that facial expressions are not just emotional expressions, but that the afferent sensory feedback from the facial action can also influence the emotional experience [3]. Therefore, related resources such as EMFACS (emotional FACS), the FACS Investigators' Guide [43] as well as the FACS interpretive database [46,47] have been elaborated to make emotion-based inferences from single and/or combinations of AUs. As suggested in the FACS Investigators' Guide [43], it is possible to map AUs onto the basic emotion categories using a finite number of rules, as we will explain later in this paper.

### 3. Research Objective

Nonverbal cues such as facial expressions, body language, and eye-contact have a key role in human communication and send powerful indications about a person's cognitive and emotional state. There are many ways that these signals may be used in order to build trust or to put people at ease. They can also be used to offend, confuse, and undermine what the speaker is trying to convey [48]. For instance, facial expressions give virtual agents the ability to communicate with users at a more emotional level, enhancing their social abilities [49]. Hence, the development of an embodied conversational agent capable of exhibiting a humanlike behavior requires incorporating well-designed non-verbal communication signals into the agent's dialogue. Besides, while a variety of approaches have been developed for the purpose of achieving such multimodal computer-assisted conversational agents, it is important to bear in mind that less research has been conducted on the development of conversational agents that can exhibit empathetic non-verbal listening behaviors. Furthermore, a clear indication of the potential for these nonverbal cues to foster language production is lacking in the literature. It follows that our interest in this study is rooted in the emotional processes affecting learners' motivation towards L2 production and is intrinsically motivated by the need to investigate the potential of emotionally expressive conversational agents towards fostering language production by second-language learners.

An independent speaking task, for instance, is a story-like communication task in which second-language learners are expected to express their opinions or ideas clearly and concisely on a particular topic within a limited amount of time, requiring them to describe their arguments clearly. However, despite their linguistic abilities, some learners may struggle to perform up to their full potential due to the amount of stress involved with such a resource-intensive activity regardless of their ability to speak the language. Considering a virtual human agent capable of displaying empathetic listening behavior as a promising approach towards resolving such issues, we are interested in its potential to assist L2 learners in overcoming their fear of failing, gaining confidence, and communicating effectively during independent speaking tasks. In consideration of the above, the goal of this study, which is an extension of our previous paper [50], is two-fold:

- Propose a method for achieving emotionally expressive computer-based agents that could display attentive nonverbal signals while listening to human conversation partners.
- Investigate the extent to which such agents are capable of conveying sufficient empathy to regulate second language learners' emotional experience and promote their production of the target language.

### 4. Approach

In order to accomplish the aforementioned research goals, we propose a method for enabling virtual human agents to display believable nonverbal listening behavior by utilizing three kinds of nonverbal signals: facial expressions, nodding, and gaze. We employed a digital human agent developed by Trulience [51], which offers highly realistic, interactive animated avatars that will help facilitate the high level of expressivity required in the context of this work.

#### 4.1. Virtual Agent's Feedback-Generation Module

To achieve attentive listening behavior, virtual human agents need to demonstrate believable nonverbal feedbacks in accordance with the current interaction status. In the present study, we aimed to simulate these attentive listening behaviors by equipping a virtual human with the ability to convey specific emotions to a listener through the use of facial expressions, nodding, and gazing movements. To this end, the virtual agent's facial expressions and movements were designed and coded in a reliable manner using the Facial Action Coding System (FACS) [6].

As described earlier, all visually discernible facial movements can be described using FACS, which breaks down facial expressions into individual components of muscle movement, called Action Units (AUs). Moreover, targeting such AUs could be especially quite

interesting because it is believed that facial expressions used to display basic emotions, such as happiness, sadness, anger, surprise, fear, and disgust, are universal and can be categorized as combinations of a variety of AUs. Therefore, when we conducted this study, we followed the method described in [52] closely and carefully categorized all of our virtual agents' face blendshapes to ensure we targeted the 64 AUs identified by Ekman. Later in this paper, we illustrate how we enable a virtual agent to display each of Ekman's six fundamental universal emotions along with nodding and gazing behavior by combining some specific AUs. For example, surprise is generated by combining Action Unit 1 (Inner Brow Raiser), Action Unit 2 (Outer Brow Raiser), Action Unit 5 (Upper Lid Raiser), and Action Unit 26 (Jaw Drop). In contrast, both gazing and nodding behaviors are generated either from a single pair of AUs or from a combination of two or more different AUs. As shown in Figure 1, our implemented virtual human agent, Truly, can display a variety of facial expressions in addition to nodding and gazing. Table 1 shows a list of the target morphs that are used to generate these non-verbal emotional feedbacks.

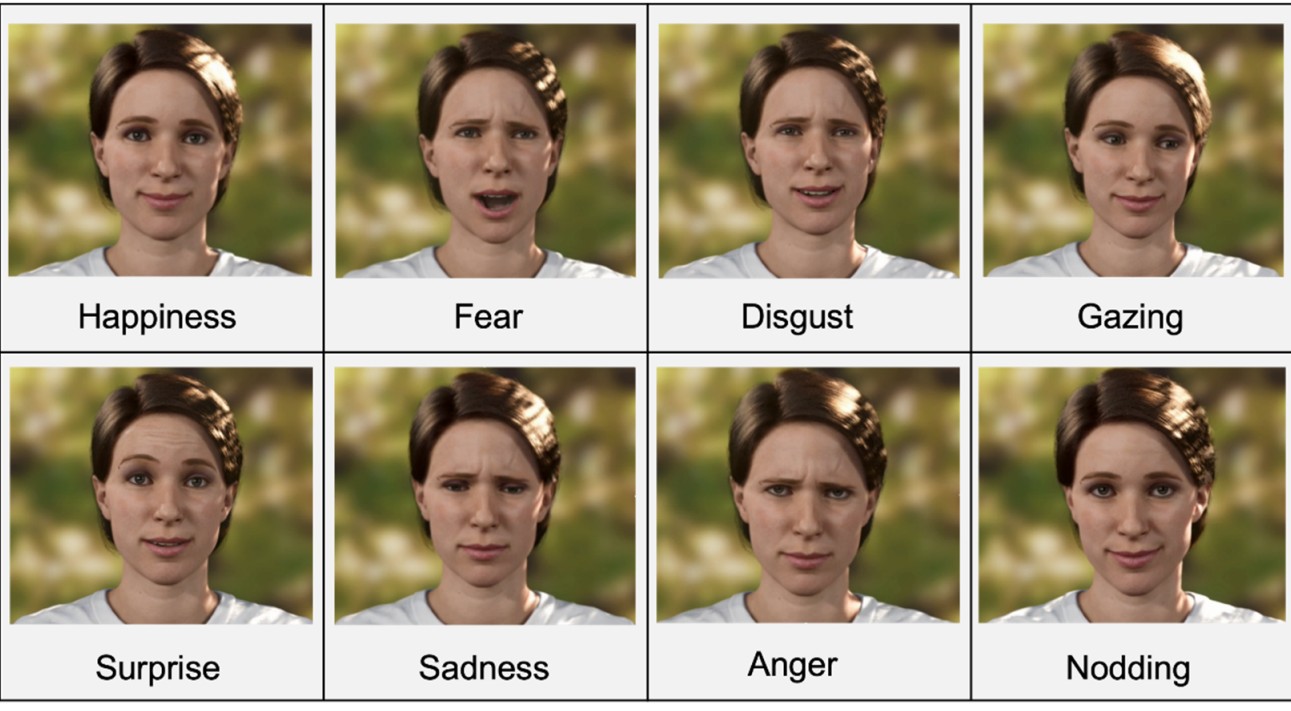

**Figure 1.** Various facial expressions displayed by the virtual agent, Truly: they include Ekman's six basic emotions in addition to gazing and nodding.

Virtual agents must be endowed with the ability to display facial expressions that align with the conversation's content in order to display particular attentive listening behaviors at a given point in an interaction in a natural way. As this requires real-time processing and understanding of the current interaction state, it is not easy to accomplish. In other terms, to convey empathetic and natural listening behavior, a virtual agent's facial expression needs to match the content of the interlocutor's speech. For example, a desirable virtual agent should display some facial feedback that conveys happiness when his or her interlocutor is discussing a happy or fun story.

As we were first eager to clarify whether nonverbal emotional feedbacks displayed by a virtual agent could have a positive impact on learners' production of the target language, we decided to adopt a cost effective and easily implementable approach. Thus, we decided to implement a system so that a wizard (i.e., a human being that partially operates the agent behind the scenes) is used to determine the best timing and type of feedbacks to display to the learner, as shown in Figure 2. In the next section, we describe how the wizard's face expressions are used to trigger the virtual agent's feedbacks.

**Table 1.** Overview of major morph targets associated with Truli's emotional feedbacks.

| | | | | |
|---|---|---|---|---|
| 1. Left_EyesLidsSquint | 2. Left_EyesLidsLookLeft | 3. Left_BrowsIn | 4. Left_NoseWrinkler | 5. All_SmileLeft |
| 6. Right_EyesLidsSquint | 7. Right_EyesLidsLookLeft | 8. Right_BrowsIn | 9. Right_NoseWrinkler | 10. Left_SmileLispOpenLeft |
| 11. Left_EyesLidsCloseHard | 12. Left_EyesLidsLookUp | 13. Left_BrowsInRaised | 14. Left_NoseScrunch | 15. All_SmileRight |
| 16. Right_EyesLidsCloseHard | 17. Right_EyesLidsLookUp | 18. Right_BrowsInRaised | 19. Right_NoseScrunch | 20. Left_SmileLispOpenRight |
| 21. Left_EyesLidsScrunch | 22. Left_EyesLidsLookUpLeft | 23. Left_BrowsDownScrunchEyes | 24. Left_NoseSneer | 25. Left_Frown |
| 26. Right_EyesLidsScrunch | 27. Right_EyesLidsLookUpLeft | 28. Right_BrowsDownScrunchEyes | 29. Right_NoseSneer | 30. Right_Frown |
| 31. Left_EyesLidsBlink | 32. Left_EyesLidsLookUpRight | 33. Left_BrowsSneer | 34. Left_NoseBrowsDown | 35. Left_Kiss |
| 36. Right_EyesLidsBlink | 37. Right_EyesLidsLookUpRight | 38. Right_BrowsSneer | 39. Right_NoseBrowsDown | 40. Right_Kiss |
| 41. Left_EyesLidsHalfClosed | 42. Left_EyesLidsWide | 43. Left_CheekRaiser | 44. Left_NoseBrowsIn | 45. All_LipsLeft |
| 46. Right_EyesLidsHalfClosed | 47. Right_EyesLidsWide | 48. Right_CheekRaiser | 49. Right_NoseBrowsIn | 50. All_LipsRight |
| 51. Left_EyesLidsBlinkLowerLidRaised | 52. Left_EyesLidsCheekRaiser | 53. Left_CheekScrunch | 54. Left_NosePull | 55. Left_FunnelBigCH |
| 56. Right_EyesLidsBlinkLowerLidRaised | 57. Right_EyesLidsCheekRaiser | 58. Right_CheekScrunch | 59. Right_NosePull | 60. Right_FunnelBigCH |
| 61. Left_EyesLidsLookDown | 62. Left_EyesLidsSmile | 63. Left_CheekSmile | 64. Left_NostrilDilator | 65. Left_FunnelClosed |
| 66. Right_EyesLidsLookDown | 67. Right_EyesLidsSmile | 68. Right_CheekSmile | 69. Right_NostrilDilator | 70. Right_FunnelClosed |
| 71. Left_EyesLidsLookDownLeft | 72. Left_BrowsUp | 73. All_CheekSmileLeft | 74. Left_LipsNoseWrinkler | 75. Left_UpperLipRaiser |
| 76. Right_EyesLidsLookDownLeft | 77. Right_BrowsUp | 78. All_CheekSmileRight | 79. Right_LipsNoseWrinkler | 80. Right_UpperLipRaiser |
| 81. Left_EyesLidsLookDownRight | 82. Left_BrowsOuterUp | 83. Left_CheekSneer | 84. Left_SmileSharp | 85. Left_LowerLipDepresser |
| 86. Right_EyesLidsLookDownRight | 87. Right_BrowsOuterUp | 88. Right_CheekSneer | 89. Right_SmileSharp | 90. Right_LowerLipDepresser |
| 91. Left_EyesLidsLookRight | 92. Left_BrowsDown | 93. Left_CheekLipRaiser | 94. All_Smile | 95. Left_SneerUpperLipFunnel |
| 96. Right_EyesLidsLookRight | 97. Right_BrowsDown | 98. Right_CheekLipRaiser | 99. All_SmileLispOpen | 100. Right_SneerUpperLipFunnel |

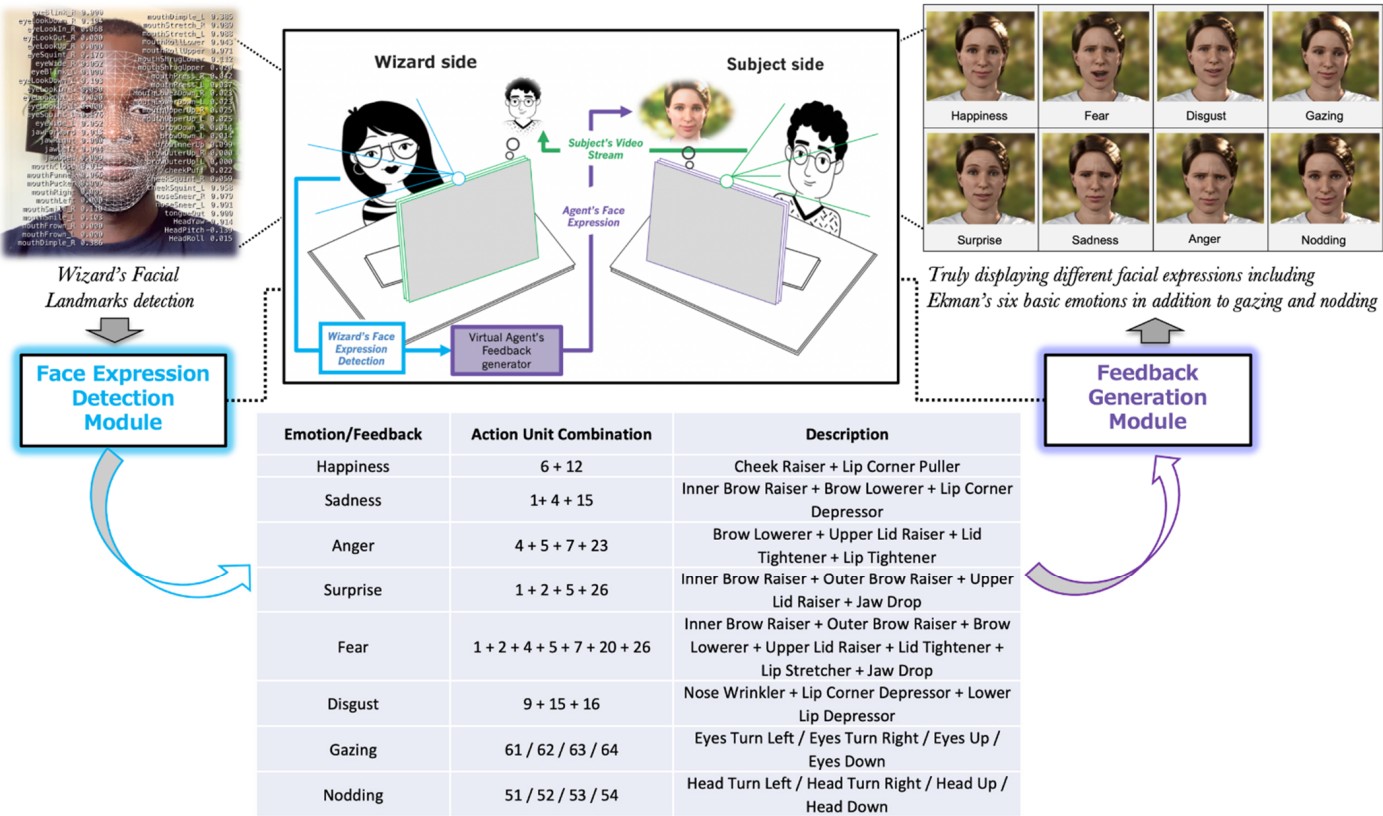

*Leveraging the Facial Action Coding System (FACS) for nonverbal cues detection and generation*

| Emotion/Feedback | Action Unit Combination | Description |
| --- | --- | --- |
| Happiness | 6 + 12 | Cheek Raiser + Lip Corner Puller |
| Sadness | 1+ 4 + 15 | Inner Brow Raiser + Brow Lowerer + Lip Corner Depressor |
| Anger | 4 + 5 + 7 + 23 | Brow Lowerer + Upper Lid Raiser + Lid Tightener + Lip Tightener |
| Surprise | 1 + 2 + 5 + 26 | Inner Brow Raiser + Outer Brow Raiser + Upper Lid Raiser + Jaw Drop |
| Fear | 1 + 2 + 4 + 5 + 7 + 20 + 26 | Inner Brow Raiser + Outer Brow Raiser + Brow Lowerer + Upper Lid Raiser + Lid Tightener + Lip Stretcher + Jaw Drop |
| Disgust | 9 + 15 + 16 | Nose Wrinkler + Lip Corner Depressor + Lower Lip Depressor |
| Gazing | 61 / 62 / 63 / 64 | Eyes Turn Left / Eyes Turn Right / Eyes Up / Eyes Down |
| Nodding | 51 / 52 / 53 / 54 | Head Turn Left / Head Turn Right / Head Up / Head Down |

**Figure 2.** Overview of the proposed WoZ experimental environment. The wizard's facial expressions are detected via the face expression detection module (left-hand side). Based on the detected expression, the virtual agent's facial expression is generated via the feedback generation module (right-hand side) by combining several action units as specified in the FACS (center).

### 4.2. Wizard's Face Expression Detection Module

A third-party API, face-api.js [53], was used to detect the wizard's face expressions in real time.

It is a JavaScript API for face detection and face recognition in the browser implemented on top of the tensorflow.js core API. Despite its lightweight design, the face expression recognition model of face-api.js is fast and provides reasonable accuracy. A depthwise separable convolution and densely connected blocks are used in the model, which weighs in at approximately 310 kb. It has been trained on a variety of images from publicly available datasets as well as images scraped from the web, according to the developer.

For Nodding and Gazing behavior detection, we used face-api.js's face recognition model, which implements a very lightweight and fast, yet accurate, 68-point face landmark detector. The default model has a size of only 350 kb (face_landmark_68_model) and the tiny model is only 80 kb (face_landmark_68_tiny_model). Both models employ the ideas of depthwise separable convolutions as well as densely connected blocks. The models have been trained on a dataset of ~35 k face images labeled with 68 face landmark point, according to the developer. Figure 3 shows a template image for landmark detection using the 68 point for frontal image [54].

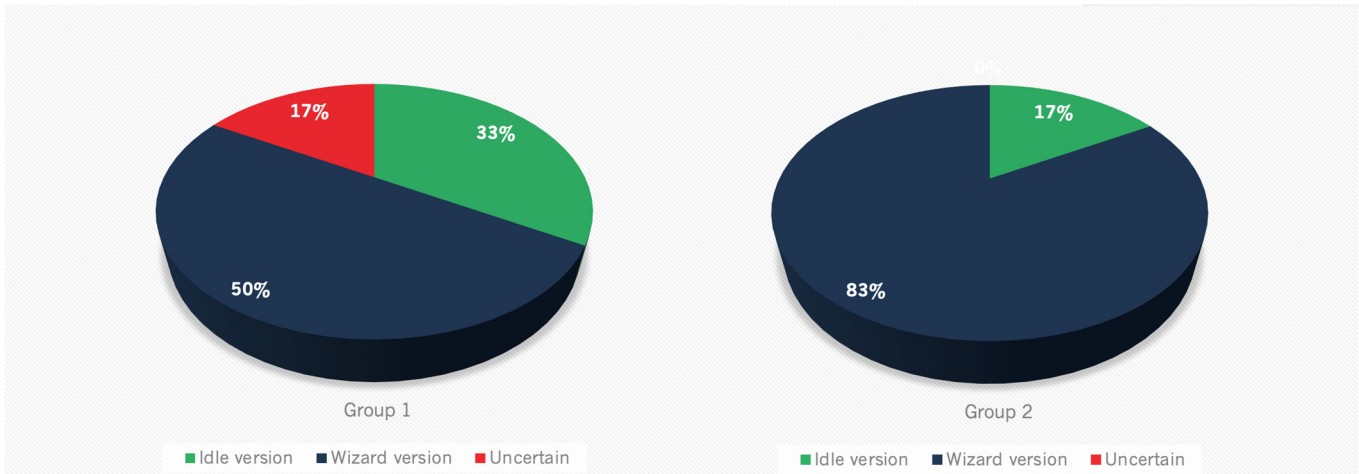

**Figure 3.** Preference questionnaire results showing ratio of subjects that preferred each version of the proposed system in each of the two groups.

A combination of appropriate action units was used to create the virtual agent's nonverbal signals based on the detected wizard's face expression. In this setting, we designed an experimental environment in which subjects (second-language learners) were unaware that they were interacting with a wizard-operated virtual agent.

## 5. Woz Experiment

### 5.1. Instruments and Participants

To evaluate the meaningfulness of the proposed system, we adopted a Wizard of Oz (Woz) experiment style [55] and implemented a prototype system where ideal timing and feedback types are indirectly selected by the wizard (i.e., a person who partially operates the agent's facial expressions behind the scenes), as shown in Figure 2. We implemented the system employing a web application architecture (Nodejs) where both parties (i.e., the wizard and the learner) communicated through a dedicated server. The wizard had on his screen the video stream coming from the learner, while on the other hand, the learner had at his end the virtual agent's video stream without knowing that the virtual agent's nonverbal feedbacks were indirectly triggered by the wizard.

In order to evaluate the meaningfulness of the proposed system, we conducted an experimental evaluation of the proposed system, in which the virtual agent interacted with a human subject via nonverbal feedbacks. Within this scenario, the human subjects, who are second-language learners, were presented with an independent speaking task such as "Describe any one of the best moments of your life". They were instructed to think and formulate their answer within 5 min while talking to the virtual agent. We prepared two versions of the system, one in which non-verbal feedbacks were indirectly triggered by the wizard and one in which the virtual agent was just in an idle state and did not provide any particular facial feedbacks on purpose. Twelve Japanese undergraduates and graduate students attending a Japanese university were recruited to evaluate the system. Participants' linguistic backgrounds were fairly homogeneous; all were native Japanese speakers and none had lived outside the country. The study's participants were informed they could withdraw from it at any time, and the results would remain anonymous.

### 5.2. Flow of Interactions

The evaluation was conducted in four steps, as listed in Table 2. Note that in order to ensure the homogeneity of experimental conditions, the same wizard was used for all interactions with learners. Also, we had to carry out the experiment over several days since learners participated in the experiment alone at different time slots so that the wizard could have enough time to rest between each round of interactions. After initial guidance,

through which they were given a brief overview of the experiment, participants interacted with each of the two system versions. Then, they were asked to fill in a questionnaire survey about the naturalness of the virtual agent's listening attitude. They were also asked to choose which of their two interactions (i.e., which version of the system) they preferred the most and the reason for their choice. Note that in Step 2 and 3, learners were presented with a different independent speaking task.

**Table 2.** Overview of the experimental flow.

|        | Group 1 (n = 6) | Group 2 (n = 6) |
| ------ | --------------- | --------------- |
| Step 1 | Initial Guidance | |
| Step 2 | Wizard condition[1] | Idle condition |
| Step 3 | Idle condition[2] | Wizard condition |
| Step 4 | Preference Survey | |

The counterbalancing method proposed by Howitt and Cramer [56] was applied to the learners' interactions with the system in each group in order to minimize the possibility that their preferences would be only influenced by their interactions with different versions of the system (i.e., order effect). To this extent, participants of Group 1 first interacted with the Wizard enabled system while those of Group 2 started with the Idle system (Step 2). In step 3, those of Group 1 interacted with the Idle system while those of Group 2 were presented with the wizard version, as shown in Table 2.

Moreover, interaction time logs and transcripts from all interactions in each condition were saved for the purpose of analyzing learners' output afterwards.

*5.3. Results*

The preference rate of the wizard-enabled version was uniformly high across the two groups, as shown in Figure 3; this version was preferred by 8 participants out of 12 (67%) in total, whereas the idle version was preferred by 3 participants out of 12 (25%). One participant declared that he did not have any preference. Note that in both groups, no matter the order in which learners interacted with the system, most of them favored the system which displayed nonverbal emotional feedbacks. In fact, among reasons justifying their choices, participants who preferred the wizard version frequently mentioned that they found the way the virtual agent Truly reacted to their talk to be encouraging, especially when they were struggling to find appropriate words to express what they were trying to say.

The above result was also corroborated by the number of words produced by learners during their interactions with the system, as shown in Figure 4. For instance, we found that participants tended to produce more words on average (M = 75.86, SD:15.22) when interacting with the wizard version compared to the number of words produced when using the idle version (M = 50.66, SD = 20.35). Interestingly, we were not able to find a clear difference regarding amount of time spent on the task in both conditions. This is probably due to the 5 min time constraint that was set for the independent speaking task.

*5.4. Discussion*

The above results allow us to draw a number of preliminary conclusions. First, the nonverbal feedbacks displayed by the virtual agent proved to be promising in motivating L2 learners towards the production of the target language, much more than a version of the system showing the virtual agent in an idle state. This is corroborated by both the results of the interactions transcripts analysis and those of the preference survey, confirming our initial beliefs that nonverbal emotional feedbacks may play an important role towards fostering virtual agents' ability to effectively trigger second-language production. Thus, we feel that these results enable us to tentatively conclude that the participants in this study will certainly display higher willingness to output the target language if given opportunities to interact frequently with this kind of system. As motivation seems to influence learners'

actual use frequency of the target language [11], it is crucial to create environments that encourage their motivation to produce the language. As it stands, it appears that the virtual agent environment proposed in this study offered quite a few benefits for the participants in the study in terms of encouraging their willingness to produce the language.

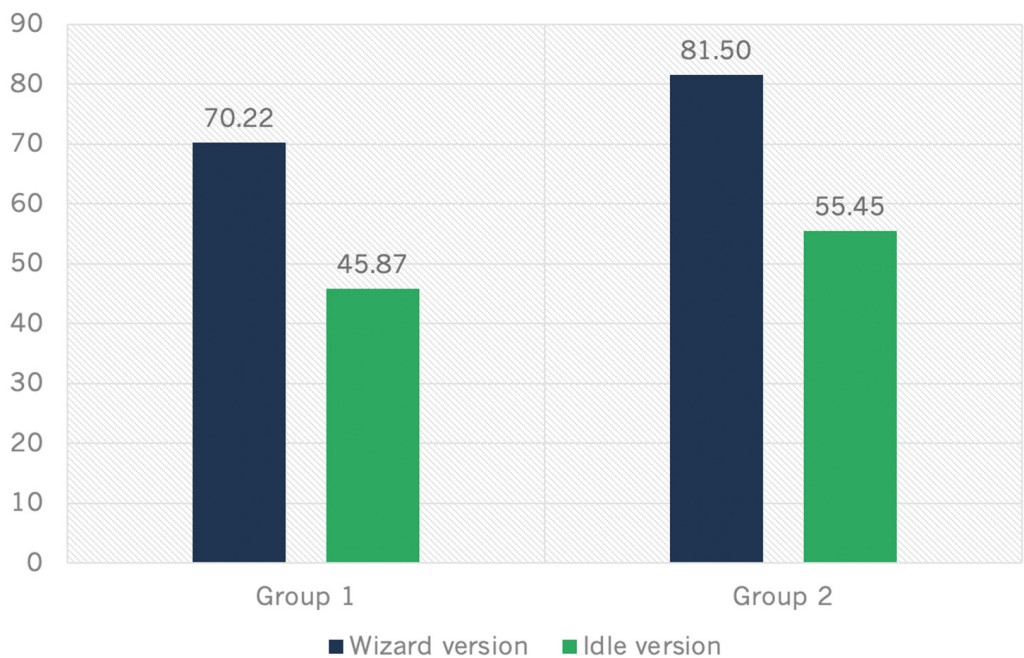

**Figure 4.** Number of words produced during speaking task by subjects in each of the two groups. Wizard condition indicates an experimental setting where the virtual agent nonverbal feedback was triggered by the wizard. Idle condition indicates an experimental setting where the virtual agent is in an idle state and does not provide any nonverbal feedbacks.

The media equation [57] that claims that humans tend to respond to media, including computer artifacts, as if they were real, is presumably one of the foundations for relation-building between learners and computer-based agents. The insights presented above tend to confirm that mimicking a wizard's nonverbal feedbacks might help identify which social rules and habits could be targeted to generate agents that display emotionally appropriate attention patterns towards second-language learners.

In sum, we feel that the present work contributes to a challenge to the computer-supported language-learning research community: the need to look beyond cognition and investigate ways to implement cost-effective approaches towards restoring the balance between cognition and emotion within embodied conversational systems. The achievement of such ambition is obviously not going to be straightforward, given the immensely diverse and complex set of phenomena related to emotion and the huge variability of learners and contexts within which they might need assistance. Innovative approaches are required to address the various empirical and theoretical questions that still need to be answered to develop caring learning support systems that were envisioned by Self [58], as systems to enhance the "whole learner" as someone sensitive to, and able to, regulate cognitive, affective, and motivational aspects of their learning.

Although the above could be seen as preliminary evidence to confirm the effectiveness of the proposed system, still, we are aware that our work presents some limitations.

First, it might be important to keep in mind that reactions from a human teacher or human peer do not always have the same effect as "identical" reactions delivered from computerized systems [59]. Hence, whether our approach, which consisted of pursuing more naturalistic display of emotion by animated agents, is desirable remains somewhat of an open question that future research should address in much more detail. Besides, the relatively modest sample size, as well as the Woz approach employed here, prevents

us from reasonably generalizing our findings without taking the risk of overestimating the significance of our results. Nevertheless, it seems important to bear in mind that while various theories related to affect and motivation have been proposed in the areas of psychology and cognitive sciences, the range of working operational models of such theories exploitable by computer-based learning support systems is very limited. This obviously highlights a need for more work towards identifying which key attributes of computer systems, and particularly embodied conversational agents, are able to influence learners' emotions and create friendly and engaging learning environments. We feel that the present work is consistent with such a view.

## 6. Conclusions

Whether we are examining race, ethnicity, culture, gender, religion, or age, it is well-known that emotional expression knows no boundaries. However, it remains to be demonstrated whether a computer agent with emotional expression would prove useful for human communication. In addition, aspects of language learning, such as emotion and motivation, which have traditionally received less attention are undoubtedly a promising research avenue, given the complementary relation that exists between such constructs and cognition. In this research, we designed a system that attempts to evaluate the extent to which the nonverbal emotional signals displayed by a virtual agent could induce positive emotions and thereby facilitating learners' production of a target language. The trends that we observed through the evaluation of the proposed system pointed towards the meaningfulness of our approach.

Our future work will focus on conducting a large-scale, in-depth evaluation of the system. As part of this effort, we will also work towards utilizing the learner's verbal input and their facial expressions to enable the virtual agent to autonomously respond in an emotionally meaningful manner. Despite the increasing interest in giving virtual humans characteristics such as affect, personality, and the ability to interact with others, little consideration has been given to how these characteristics can modulate the empathetic behavior of these virtual beings. Therefore, we believe that manipulating a virtual agent's empathic behavior by adjusting its mood, personality, and relationship to its interaction partner can be helpful in generating more appropriate attentive listening behaviors.

**Author Contributions:** Conceptualization, methodology writing—original draft preparation, funding acquisition E.A.; supervision, review and editing M.T. All authors have read and agreed to the published version of the manuscript.

**Funding:** This research was supported by JSPS KAKENHI Grant Number #22K18011.

**Informed Consent Statement:** Informed consent was obtained from all subjects involved in the study.

**Data Availability Statement:** All of the data is contained within the article.

**Conflicts of Interest:** The authors declare no conflict of interest.

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
