# Peer review of "Towards Emotionally Expressive Virtual Human Agents to Foster L2 Production: Insights from a Preliminary Woz Experiment"

_mti, doi:10.3390/mti6090077_

Round 1

Reviewer 1 Report

The study focuses on VA's emotional feedback in second language communication. In particular, the aim of the study is to determine whether and to which extent emotional expressions reproduced by virtual agents could provide empathetic support to second language learners during communication tasks. For this purpose authors described a virtual agent that can display nonverbal signals that include emotions from facial expressions and gazing and head movement; they also evaluated their efforts. From a theoretical point of view, the work is well based even if considering that virtual agents can have a relationship with their user's authors can include in their considerations also the notion of VA 'stance': for instance ‘dominant/assertive or humble one. For this purpose, authors can, in the limitations, add something about this aspect, considering the following works:

Nass, C., Moon, Y., Fogg, B. J., Reeves, B., & Dryer, D. C. (1995). Can computer personalities be human personalities?. International Journal of Human-Computer Studies, 43(2), 223-239. 

D’Errico, F., & Poggi, I. (2019, January). Tracking a leader’s humility and its emotions from body, face and voice. In Web Intelligence (Vol. 17, No. 1, pp. 63-74). IOS Press.

Another limitation is the weak number of participants in this first evaluation that needs to be enlarged, but considering that the VA is innovative from an applicative point of view, it can be considered for publication. 

Author Response

We would like to thank the reviewer for his kind and insightful comments that were helpful to improve the quality of our paper.  The following changes were made to address the reviewer’s evaluations, comments and suggestions:
We mentioned the suggested works by the reviewer in the last paragraph of section 4.4 Discussion.
We also acknowledged the sample size issue in the fifth paragraph of section 4.4 Discussion.
We added figures 3 and 4 to improve the readability of obtained results in section 4.3 Results.

Reviewer 2 Report

This paper presents a novel system to evaluate the extent to which the nonverbal emotional signals displayed by a virtual agent could induce positive emotion to facilitate foreign language learners' oral production.

The authors' idea, which is technically sound, is to aid foreign language learners with a virtual study buddy. This is an embodied conversational agent. Thanks to facial expression, based on the FACS system, it is possible to provide empathetic feedback to the learners. A theoretical justification of this approach based on literature is reported in the paper.
The proposal has been validated by a "Wizard of Oz" experiment where a human (the wizard) reacts to the learner's speech by means of his/her facial expression has been set. His/her facial expressions are detected in real-time by a facial landmark detection algorithm and then reproduced in computer graphics using a FACS-compliant visualizer (Truly). The wizard can listen to and see a video stream of the learner’s face, while the learner can only see the computer graphic representation.
The system, of course, is not complete since the loop is closed by a human, but the authors want to continue this study by increasing the number of learner testers, and then after statistically significant results are obtained, if it demonstrates an effective, closing the loop, substituting the human by a machine learning algorithm.

The paper title contains the keywords “towards” and "preliminary" representing the preliminary content of the paper honestly.
Technical reproducibility of the test is possible given the description of the methodology provided in the paper.

Author Response

We would like to extend our deep appreciation to the reviewer for understanding and highlighting the meaningfulness of this work. We will continue to work towards ultimately closing the loop by a machine learning algorithm, as suggested by the reviewer.  We mentioned this in the fifth paragraph of section 4.4 Discussion.

Round 2

Reviewer 1 Report

All comments are addressed. You can accept this paper.